# Using Artificial Neural Networks to Assess Earthquake Vulnerability in Urban Blocks of Tehran

**Rasoul Afsari [1,\*], Saman Nadizadeh Shorabeh [2]** **, Amir Reza Bakhshi Lomer [3], Mehdi Homaee [4,†]** **and Jamal Jokar Arsanjani [5]**

[1] Department of Passive Defense (Urban Planning of Passive Defense), Superme National Defense University, Tehran 1698613411, Iran
[2] Department of GIS and Remote Sensing, Faculty of Geography, University of Tehran, Tehran 1417935840, Iran
[3] Department of Geography, Birkbeck, University of London, London WC1E 7HX, UK
[4] Department of Mining and Environmental Engineering, Faculty of Engineering, Tarbiat Modares University, Tehran 14115, Iran
[5] Geoinformatics Research Group, Department of Planning and Development, Aalborg University Copenhagen, 2450 Copenhagen, Denmark
[\*] Correspondence: r.afsari@sndu.ac.ir
[†] Agrohydrology Research Group (Grant No. IG-39713), Tarbiat Modares University, Tehran 14115, Iran.

**Abstract:** The purpose of this study is to assess the vulnerability of urban blocks to earthquakes for Tehran as a city built on geological faults using an artificial neural network—multi-layer perceptron (ANN-MLP). Therefore, we first classified earthquake vulnerability evaluation criteria into three categories: exposure, sensitivity, and adaptability capacity attributed to a total of 16 spatial criteria, which were inputted into the neural network. To train the neural network and compute an earthquake vulnerability map, we used a combined Multi-Criteria Decision Analysis (MCDA) process with 167 vulnerable locations as training data, of which 70% (117 points) were used for training, and 30% (50 points) were used for testing and validation. The Mean Average Error (*MAE*) of the implemented neural network was 0.085, which proves the efficacy of the designed model. The results showed that 29% of Tehran's total area is extremely vulnerable to earthquakes. Our factor importance analysis showed that factors such as proximity to fault lines, high population density, and environmental factors gained higher importance scores for earthquake vulnerability assessment of the given case study. This methodical approach and the choice of data and methods can provide insight into scaling up the study to other regions. In addition, the resultant outcomes can help decision makers and relevant stakeholders to mitigate risks through resilience building.

**Keywords:** vulnerability; earthquake; risk assessment; artificial neural networks; Tehran

## 1. Introduction

Earthquakes are considered one of the most devastating natural disasters causing severe physical, social, and financial damage [1]. Nearly two million people were harmed by earthquakes in the 20th century [2]. The goal of urban planning is to significantly reduce the effects of natural disasters and increase safety [3]. As a result of indiscriminate development, improper planning and design, and structural failure, human progress has been limited in developing countries to deal with earthquake-related challenges [4]. Many people were killed, and significant economic losses were caused by the earthquake [5]. Since 1900, Iran has suffered more than one million injuries [6], and more than 180 thousand people have been killed in the past five decades [7]. The earthquake vulnerability index of Iran is among the most dangerous in the world, which measures the amount of damage that can be caused by different magnitude earthquakes [7–10]. In this regard, proper and timely preparation can reduce human and financial losses if the catastrophic effects of an earthquake are calculated in advance.

Old cities were built at a time when human knowledge of geological faults and potential earthquakes was limited. Rapid urbanization around these cities and the need for extra housing for the growing population have resulted in a lack of inspections before building construction. This is particularly true in developing countries [11]. We have learned an alarming lesson from the recent devastating earthquakes in Turkey and Syria. This is why there is a need to screen cities for their vulnerability to earthquakes and the risks associated with them. We perform such studies to take preventive measures, such as relocating the population and upgrading the infrastructure.

This can become even more problematic as there is a lack of proper actions to revise the urban development plans that a hypothetical earthquake is projected [12]. It is, therefore, necessary to develop robust approaches and state-of-the-art modeling techniques, such as artificial intelligence, to screen earthquake vulnerability. By combining spatial factors such as exposure, sensitivity, and adaptive capacity into a multi-criteria decision-making framework, combined with machine learning methods in a GIS setting, this study seeks to provide this information. We need to consider a process that combines and transforms spatial data (metric maps) and values associated with people's judgment (priority of decision-makers) in order to obtain valuable information for decision-making [13,14]. GIS, on the other hand, is a valuable tool for storing, manipulating, analyzing, and managing spatial data [15,16]. Consequently, by integrating GIS and MCDA-ML methods, decision makers would be able to perform decision analysis functions such as ranking options to select the appropriate area [17,18].

Artificial Neural Networks (ANNs) inspired by the biological neural system for information processing are one of the most popular machine learning techniques used in environmental studies in recent years [19]. ANNs have a large number of highly interconnected processing elements (neurons) that work in concert to solve specific problems [20,21]. Compared to traditional statistical models, ANNs are independent of the statistical distribution of data. ANNs do not need prior knowledge of data to extract patterns [22]. ANNs have a self-learning capability that allows them to optimize themselves over a large number of iterations. Moreover, such models delineate the potential relationship between the input layers and the given outcomes by cross-comparing the introduced ground truth data, i.e., training data, to them [23]. ANN can also determine complex patterns among data sets for which mathematical formulas are not suitable. In addition, it can handle missing and uncertain data [24,25]. Hence, ANN can serve as a suitable choice for computing earthquake vulnerability maps. In order to implement ANN using effective indicators, the network needs to be trained, which is necessary to choose the appropriate training parameters [26]. Finding the ideal and optimal network should be performed using accuracy metrics alongside an expert's interpretation [27].

The Analytic Network Process (ANP) facilitates the decision-making process by providing a structure for organizing different criteria and evaluating the importance of each one compared to alternatives [28]. The purpose of this method is to determine the weight of decision criteria using pairwise comparison [29]. The ANP method shows the complex relationships between different decision levels in a network form and considers the interactions and feedback between criteria and alternatives [30]. The weights resulting from the causal relationships between the elements, along with the internal weights of each cluster, form an initial super matrix. This super matrix is linearly weighted, and finally, the final weights of the elements are obtained by using the limit form of the weighted super matrix [31].

Various studies have investigated the vulnerability caused by earthquakes by employing a combination of geographic information systems and decision-making methods. The most relevant studies are discussed as follows. In research in Mymensingh, Bangladesh, Alam and Haque [32] evaluated the earthquake vulnerability of residential areas using an approach based on spatial multi-criteria analysis. In this study, a total of 23 spatial criteria were used in the four dimensions of geological, systematic, structural and socio-economic criteria. Hierarchical analysis method was implemented to calculate the weight of the

criteria, and the weighted linear combination method was used to prepare the vulnerability map. The findings concluded that out of 241 residential neighborhoods in Mymensingh city, 51 neighborhoods are highly vulnerable. Yavuz Kumlu and Tüdeş [33] determined the high-risk areas for earthquakes in Yalova city center (Marmara Region, Turkey) using GIS-based multi-criteria decision-making techniques. The criteria were considered in two categories: geology (geology, lithology, and liquefaction) and infrastructure (health centers, access, quality of materials, and number of floors). They concluded that the city center, located on the east side of the river, has the highest risk of earthquake due to the presence of low-quality, tall-rise, and attached buildings. Khedmatzadeh et al. [34] investigated the analysis of urban vulnerability indicators with the approach of earthquake crisis management in Urmia city. They used a total of nine spatial criteria, including population density, worn texture, type of materials, the width of roads, slope, land use, number of households in the block, type of skeleton, and infrastructure of the building. The FAHP method has been used to produce the vulnerability map. They concluded that the highest vulnerability is observed in areas with slopes greater than 20% and areas with high population density. Heydarifar and Mahmoudi [35] analyzed the vulnerability of urban land use against earthquakes in research they conducted in the city of Javanroud. They used a total of 11 location criteria in three dimensions: neighborhood, structure, and environment. The ANP method was used to assign weights to the criteria, and the WLC method was used to prepare the vulnerability map. Their results showed that about 20% of the built-up areas of Javanrood have moderate and high vulnerability against this type of hazard. Alizadeh et al. [36] presented a new hybrid framework using multi-criteria decision-making models and machine learning to assess earthquake vulnerability using social, economic, environmental, and physical indicators. Their results showed that the south and southeast areas of Tabriz city have a low vulnerability, while the northeast areas are in the class with very high vulnerability. Jena and Pradhan [37] developed a model based on AHP-TOPSIS and ANN for vulnerability risk assessment in the northern part of Sumatra Island, Indonesia. They used 18 location indicators in two groups of possible indicators and vulnerability indicators. They concluded that the proposed model generalizes better results than the traditional models and some existing probabilistic models. Yariyan et al. [2] assessed the earthquake risk in Sanandaj city using the GIS-MCDA combination. All criteria from three groups were considered in this study: environmental, physical, and demographic. Moreover, the combination of fuzzy analytic hierarchy process (FAHP) and ANN methods has been used to calculate the weight of the criteria and prepare the suitability map. They found that the earthquake risk map is more reliable when FAHP-ANN is combined. Kalakonas and Silva [38] used the ANN method combined with GIS to model the seismic vulnerability of buildings in the Balkan region. Their results indicate a superior performance of the ANN models over traditional approaches, potentially allowing a greater reliability and accuracy in scenario and probabilistic seismic risk assessment.

Based on a review of the research background, it appears that the earthquake vulnerability assessment was intended to be based on MCDA methods and limited criteria. Due to the large contribution of human knowledge to MCDA methods, the results can be uncertain. As a result, models using artificial intelligence can significantly reduce the level of uncertainty mentioned above, thereby providing a high degree of efficiency in monitoring environmental issues. In this study, the main objective was to develop an earthquake vulnerability map based on ANNs for Tehran city. Consequently, a set of training data was created based on a combined MCDA-MCE model. A map of the earthquake vulnerability was then produced using the ANN method and a comprehensive set of effective factors. Tehran is located at the foot of the Alborz mountain range and is the capital of Iran. The existence of numerous faults around and inside this city, as well as the existence of worn-out structures, structure density, dense population, non-compliance with standards, vulnerable vital arteries, being the central hub of ministerial buildings and embassies, important economic and social centers, and inappropriate physical development, are the reason for selecting it as a study area. This study thus has some innovative features, including the

following: (1) the first application of multi-layer perceptron neural networks in Tehran; (2) the consideration of a spatial database of comprehensive and complete criteria; and (3) the first time classifying criteria into components such as sensitivity, exposure, and adaptability capacity.

## 2. Vulnerability: Concept and Mapping

The term vulnerability originates from the Latin word "vulnerare" which means "damage". Therefore, vulnerability can be expressed as "damaged capacity" [39]. The concept of vulnerability was first used in the 1970s in the field of natural hazards and has been used in various disciplines since then. The concept of vulnerability mapping has been discussed in several studies, as well as analytical approaches to vulnerability assessment and mapping to environmental changes in the context of climate change [40–47]. Several studies have also provided conceptual frameworks for conducting vulnerability assessments [48–57].

As a tool to describe socio-economic and natural systems' readiness to deal with damage or risk, vulnerability has been used in interdisciplinary studies for decades. The topic has so many resources devoted to it that it is difficult to provide a single definition that covers all of them. As one of the most well-known definitions, the United Nations proposed an international strategy for disaster reduction. The term "vulnerability" is used to describe a condition that is caused by physical, social, economic, and environmental factors or processes and that reduces the preparedness of societies for the effects of disasters [58]. Moreover, the United Nations Development Programme (UNDP) defines vulnerability as a human process or condition caused by physical, social, economic, and environmental factors that determine the probability and extent of disaster damage [59]. Vulnerability is defined as a set of conditions or processes that affect the community's readiness in the first definition; vulnerability is defined in the second definition as a set of human processes or conditions. Further, vulnerability is strongly related to the environment, context, and condition; as a result, similar consequences in different societal and economic conditions often have different consequences, which may be due to spatial differences in vulnerability [60]. In addition, vulnerability is a dynamic factor that can be considered a result of the interaction between a wide range of socio-economic factors [61]. Due to this, environmental processes may be considered a serious threat to a particular society but not as a natural disaster for a society with different characteristics.

Vulnerability can be considered as a function of three components: exposure, sensitivity, and adaptability capacity, and it can be the ability of a system to adapt one of the three aspects (exposure, sensitivity, and adaptability capacity) to abnormal conditions and external forces related to risk [44]. Therefore, vulnerability can be understood as the sensitivity of a system that is damaged by exposure to environmental and social changes due to a lack of ability to adapt [62]. Having an understanding of vulnerability and the interactions between its structure and various processes is crucial. Therefore, knowledge of the system allows optimal measures to be taken to reduce and adjust its destructive effects. Therefore, vulnerability assessment can be viewed as an effective tool to determine the sensitivity of a system. It provides managers with the necessary information to prioritize protection and management programs. Most of its outputs are solutions that reduce vulnerability by reducing threats and providing technical solutions. When using this definition of vulnerability, exposure is defined as a collection of strategies for coping with environmental, social, and economic stressors. Risk exposure refers to pressures resulting from changes in the frequency, nature, intensity, duration, and area of stress [63]. Sensitivity describes the degree to which a species or system will be influenced by external factors based on its internal characteristics. The probability of experiencing different degrees of disturbances and shocks can also be defined as sensitivity [54]. A system's ability to deal with social, economic, and environmental tensions or to resolve them is referred to as its adaptability capacity. According to Lindner et al. [64], adaptability capacity is defined as the capacity of a system to adapt to and deal with stress and tension. By adjusting the exposure and

sensitivity components, Figure 1 illustrates how adaptability capacity has an influential effect on determining vulnerability.

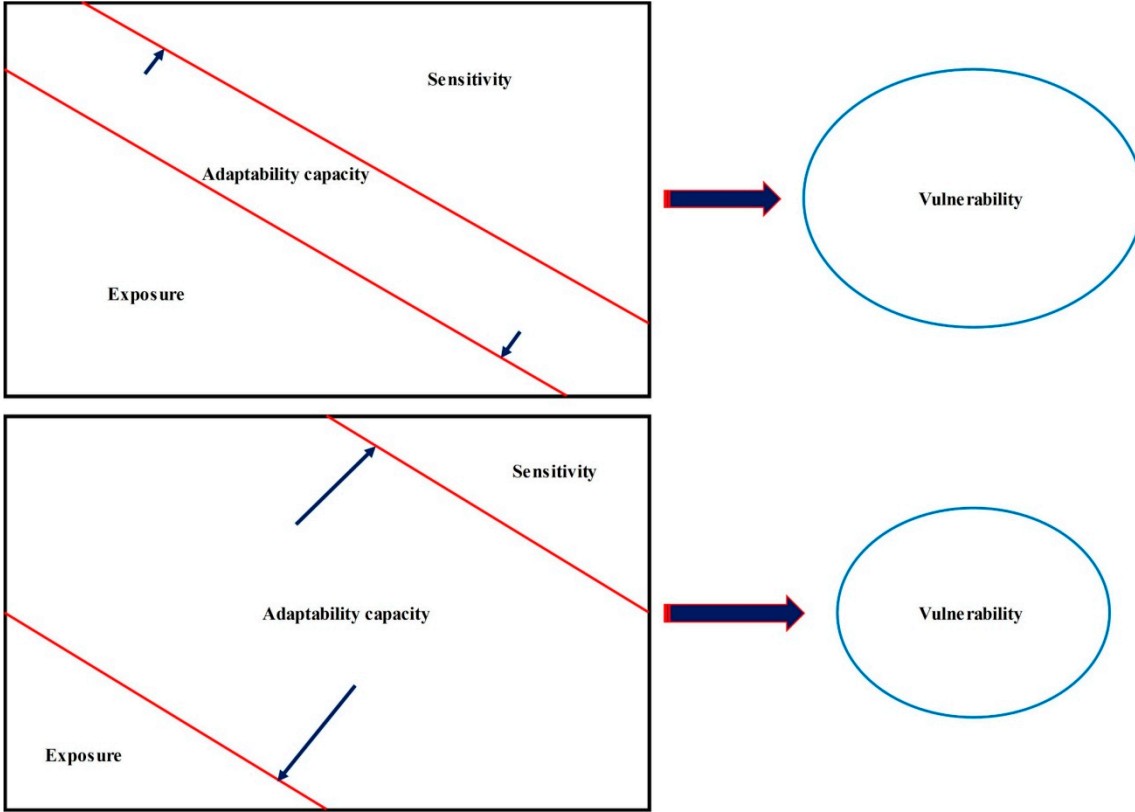

**Figure 1.** The effect of adaptability capacity on vulnerability [65].

## 3. Materials and Methods

### 3.1. Study Area and Data

The metropolis of Tehran, the capital of Iran, is the 37th most populous city in the world, with a population of 9,259,009. Tehran is geographically located from $51°17'$ to $51°33'$ east longitudes and $35°36'$ to $35°44'$ north latitudes (Figure 2). On a number of faults, including the North Tehran Fault, the North Ray Fault, and the Mosha Fault, the city of Tehran is situated on the southern foothills of the Alborz range. Alborz was formed during the Late Triassic period by the collision of Gondwana with Eurasia. Located along the Alpine–Himalayan seismic belt with a length of 600 km and a width of 100 km, it is an east–west trending mountain range. As a result of the convergence of northern central Iran and Eurasia, Alborz is a highly active tectonic region that is experiencing significant stresses from a tectonic perspective. Tehran has a high population density, rapid urban growth, and non-standard constructions that make it extremely vulnerable to earthquakes [66]. Historically, Tehran has experienced earthquakes every 150 years, and the last large earthquake occurred in 1839 AD [67]. Therefore, a major earthquake is imminent in Tehran.

Among the most prominent active faults in or around Tehran, the following can be mentioned with their main features [68,69]:

North Tehran: It is the biggest fault in Tehran, which is located south of the Alborz range. There is an east–west fault running through the north of Tehran, spanning 108 km from Lavasan and Niknamdeh (northeast of Tehran) to the west of Valian (west of Karaj) and sloping in a northward direction. The slope of the fault varies in different regions and has been measured between 10 and 80 degrees. The average sliding rate is approximately 0.3 mm year$^{-1}$.

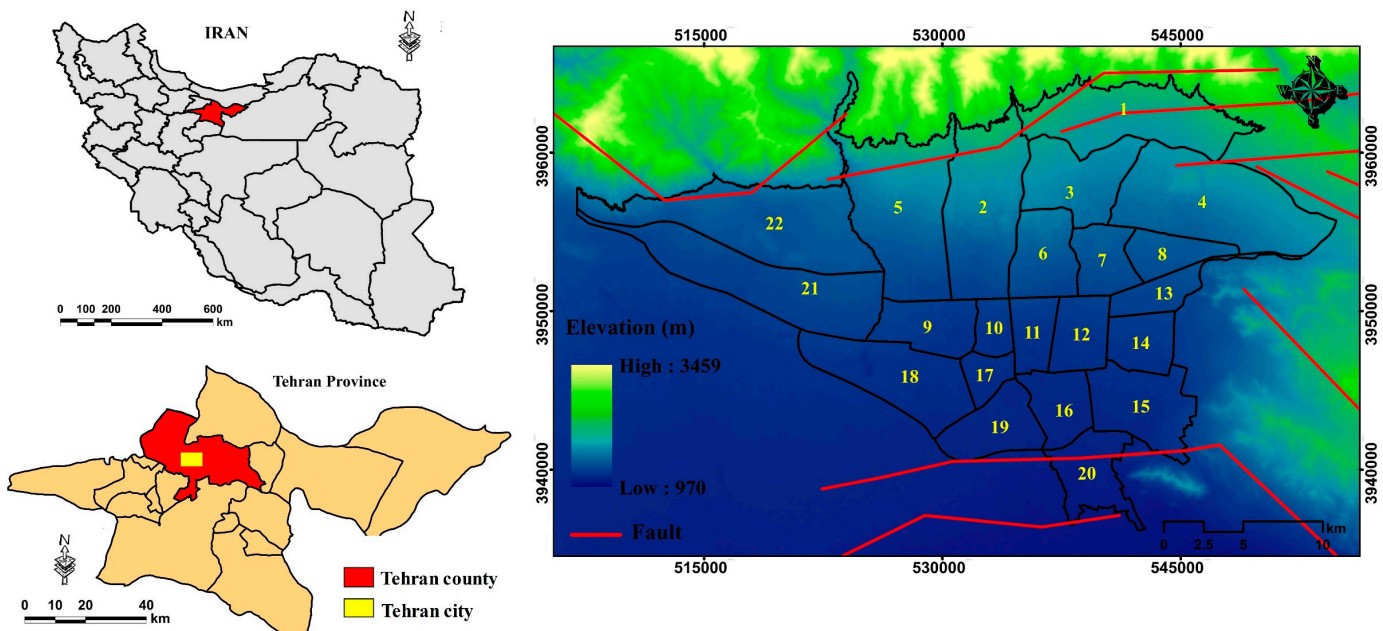

**Figure 2.** Geographical location and faults of the studied area (numbers 1 to 22 represent the urban districts of Tehran).

Mosha: It is a compressional fault with a length of 200 to 400 km. It stretches more than 10 m wide near the Mosha region and has been severely cut, crushed, and powdered. The fault has a convex shape towards the north, with slopes everywhere to the north. The slope angle varies between 35 and 70 degrees. The average slip rate is approximately 0.2 mm per year$^{-1}$.

Kahrizak-South Rey—North Rey: An area with an east–west trend located in the south of Tehran and consisting of the Kehrizak, South Ray, and North Ray faults, which have lengths of 35, 18.5, and 16.5 km, respectively.

Parchin (Ivanki): The Parchin fault has an approximate length of 80 km and is one of the central Alborz faults. It is a compressional fault that runs northwest–southeast and slopes northward, with the northern edge located in Tehran's south.

The following is a list of Iranian earthquakes near 7 Richter scales in the past and present around Tehran, as shown in Table 1. A significant earthquake occurs in the Tehran region approximately every 158 years. The last major earthquake occurred in 1830 and measured 2.7 on the Richter scale.

**Table 1.** The list of earthquakes in Iran on scales close to 7 Richter around Tehran [70].

| Year (BC) | County | Fault | Ms | MMI |
|---|---|---|---|---|
| 300 | Ray | Parchin, Ray | 7.6 | X |
| 743 | Caspian Gate | Garmsar | 7.2 | V111+ |
| 855 | Ray | kahrizak | 7.1 | V111+ |
| 958 | Teleghan | Mosha | 7.7 | X |
| 1117 | Karaj | Tehran | 7.2 | VIII+ |
| 1665 | Damavand | Mosha | 6.5 | Vi11+ |
| 1815 | Damavand | Mosha | N/A | V+ |
| 1830 | Damavand | Mosha | 7.1 | VIII+ |

Many different criteria have been examined in past studies. A number of criteria, including population density, distance from the fault, type of material, and type of skele-

ton, were used in all previous studies. The current study analyzed 16 criteria based on experts' opinions, extensive use in studies, and the availability of spatial layers, among other criteria used in previous studies. These criteria were classified into three categories, including adaptability capacity, sensitivity, and exposure. Tehran municipality provided criteria related to accessibility, such as distance from medical facilities. In addition, criteria regarding population and block characteristics were obtained from the Statistical Centre of Iran (SCI). Elevation and slope criteria were obtained from satellite images, and fault criteria were developed by the Iran National Cartographic Center. Afterward, a spatial map of the criterion was created in a GIS environment with spatial tools appropriate for each criterion. The criteria used in this study are shown in Table 2.

**Table 2.** The criteria used in this study.

| Criterion | Description |
|---|---|
| Distance from fire station | Fire stations are important and vital service centers in cities that play an important role in ensuring the safety of citizens and infrastructure [71,72]. Therefore, proximity to them will increase the efficiency of fire station services during an earthquake. |
| Distance from medical centers | Access to medical facilities (such as health centers and hospitals) plays a key role in providing services and quickly addressing the condition of affected people during and after an earthquake [73]. Therefore, convenient and quick access to medical facilities will increase resilience against earthquakes. |
| Distance from pharmacy | Pharmacies are among the important service centers in the city, and quick and timely access to them is of great importance for reducing mortality and increasing the health of injured people during an earthquake [74]. Therefore, as the distance from pharmacies increases, vulnerability increases too. |
| Density of literate population | Educated individuals can adapt to disasters more effectively and have appropriate responses during disasters due to having the necessary information and awareness about risks [36]. Therefore, a higher education level may lead to less vulnerability. |
| Working population density | Households with low job-income status do not have the necessary ability to pay for retrofitting and access to the necessary services and equipment. Therefore, strengthening and reducing vulnerability depends significantly on the employment and income status of households [75]. |
| Distance from main road | The network of urban roads is considered to be one of the most important vital arteries of the cities, which, especially after the crisis, have a significant impact on rescue operations and the evacuation of the injured [76]. Therefore, with increasing distance from the road network, vulnerability increases. |
| Distance from public transport station | Convenient access to public transportation stations will reduce traffic and prevent street closures after an earthquake. As a result, the evacuation and relocation of the affected people and the transfer of rescuers to the accident site will be faster [77]. |
| Elevation and Slope | Elevation and slope are factors affecting earthquake vulnerability in urban environments. During an earthquake, the areas located on the slope and at higher altitudes are more vulnerable to damage [78]. Moreover, serving these areas will be associated with many problems [79]. Therefore, there is a direct relationship between elevation and slope with vulnerability. |

**Table 2.** *Cont.*

| Criterion | Description |
|---|---|
| Distance from fault | Proximity to geological faults is one of the most important criteria affecting the vulnerability caused by earthquakes. Because being close to it brings great damage and vulnerability, and distance from it reduces the risk and, as a result, more resilience [80]. |
| Building quality (skeleton type and material type) | Buildings are the most important and main elements that are damaged during an earthquake [81]. Using resistant building materials and following standards in construction reduces vulnerability to earthquakes [82]. |
| Distance from fuel station | Fuel stations can create risks in the form of fire and explosion in the surrounding areas [36]; therefore, the greater the distance from them, the lower the vulnerability and vice versa. |
| Vulnerable population density | The vulnerable population includes people under 6 years old and over 60 years old. The higher the population density of vulnerable people in an area, the higher vulnerability [72,83]. |
| Total population density | In high-density areas of a city, a higher portion of the population is exposed to earthquakes, thus, higher vulnerability [84]. |
| Distance from power transmission lines | One of the important parts that are highly vulnerable due to an earthquake is the network of power transmission lines. For this reason, residential areas that are located near power transmission lines are more vulnerable than areas those farther away [85]. |

*3.2. Methodology*

A summary of the methodology process for the present study is presented in Figure 3. Based on previous studies and experts' opinions, the effective earthquake vulnerability criteria for Tehran were identified and collected from related organizations and agencies. Then, a spatial database of these criteria was created. All criteria were then divided into three categories: sensitivity, exposure, and adaptability capacity. Following this, a criteria map was prepared using spatial analysis, and a database of training locations was created using MCDA-MCE. Finally, after learning the network using training data (locations), an earthquake vulnerability map was produced by using ANN-MLP in Tehran city.

3.2.1. ANP

The weight of each criterion indicates the degree of importance of that criterion in the final decision. By changing the weight of one criterion, the degree of importance of that criterion changes in decision making. Saaty [86] introduced the ANP approach due to the inability of the analytic hierarchy process (AHP) to consider the dependencies between indicators and options. The advantage of this approach over AHP is that it considers the effective elements of decision making. The steps of the ANP model are as follows [86]:

1. Making a research network diagram: In this step, the problem should be divided into criterion levels and sub-criteria and options, if any, and the relationships between them should be identified.
2. Forming the matrix of paired comparisons: In this step, elements at each level are compared in a pairwise manner to other elements at a higher level, and matrices of paired comparisons are generated. Moreover, in the end, a pairwise comparison of internal relationships should be made.
3. Calculating the inconsistency rate: In this step, we calculate the ANP inconsistency rate. If this rate is less than 0.1, the matrix appears consistent.
4. Forming the initial super matrix: The initial super matrix is formed by using the weights of the pairwise comparisons obtained in step 2.

5.　Creation of a balanced super matrix: The balanced super matrix must be created after the initial matrix has been created.

6.　Creation of the limit super matrix: The balanced super matrix must be raised to the maximum power so that each row converges to a number, and that number is the weight of the criterion or option.

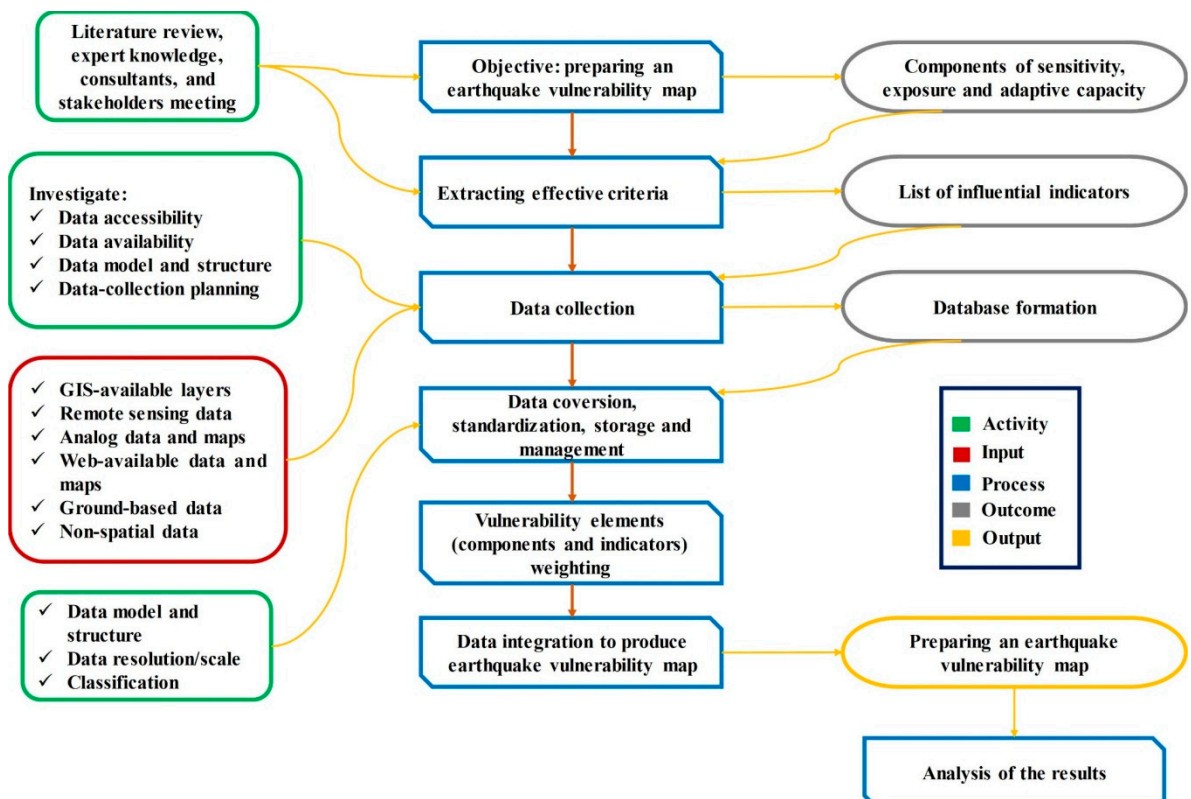

**Figure 3.** Flow diagram of study methodology.

### 3.2.2. Fuzzification

Fuzzy logic is based on "condition-result" logical rules, which utilize linguistic variables and the fuzzy decision-making process in order to depict the space of input variables into the space of output variables (using fuzzy logic) [87]. Fuzzy systems (based on logical rules) combined with ANNs (which are capable of extracting knowledge from numerical information) have been developed to provide a neural adaptive inference system [88]. To standardize system inputs, fuzzy functions are used. Fuzzy functions help to enhance the scale and distribution of data.

### 3.2.3. ANN

The ANN is inspired by a biological neural network and is made of simple computational units connected together called neurons [89]. Generally, each neural network consists of three layers, which are as follows [90]: input layer, which includes several neurons that receive the input parameters to the model in this study; hidden layer, which consists of a number of variable neurons, whose optimal number is determined by trial. Network responses are also made at the output layer (Figure 4). Therefore, each layer is built with an individual neuron that receives an input vector ($X$), performs a weighted sum ($S$), and generates a real output ($Y$) (see Equations (1) and (2)) by means of a linear or non-linear transfer function ($f$):

$$Y = f(S) \tag{1}$$

$$Y = f\left(\sum_{i=1}^{n} w_i x_i + b_i\right) \tag{2}$$

where $w = (w_1, w_2, \dots, w_n)$ are the weights of neuron $i$ (or the weight matrix), $x = (x_1, x_2, \dots, x_n)$ are the inputs of neuron $i$, $b_i$ is the bias of neuron $i$, $S$ is the weighted sum of the inputs (called the net inputs or potential of the neuron $i$), and $f$ is the transfer function (or activation function) of neuron $i$.

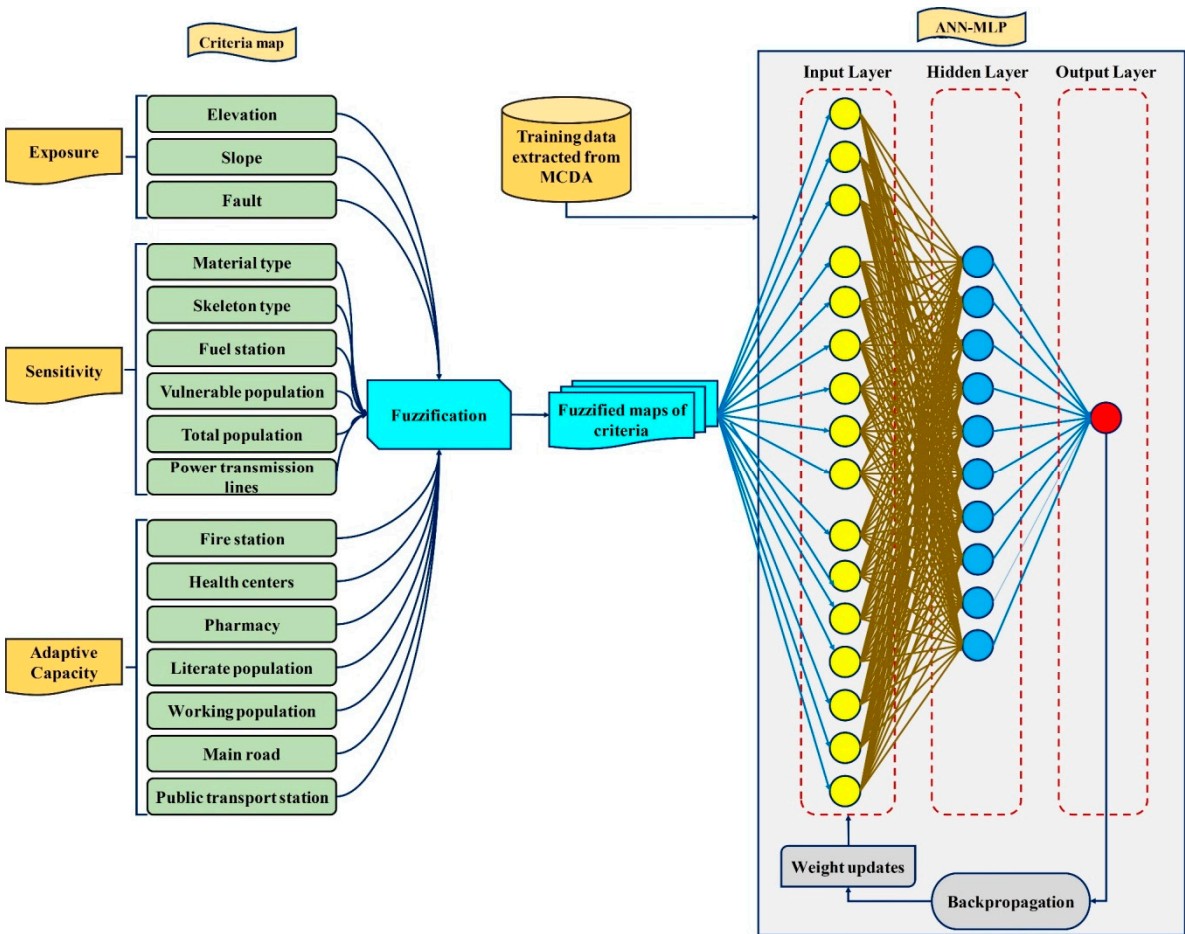

**Figure 4.** Architecture of multi-layer perceptron neural network.

There are two types of multi-layer perceptron networks: sigmoid function and sigmoid tangent. In most cases, a single neuron is not sufficient to solve problems involving a large number of inputs. In order to solve such problems, neural networks with several layers or neurons are employed [91]. Multi-layer networks have a very high degree of power. An example is a two-layer neural network consisting of a sigmoid layer and a linear layer, which can estimate any arbitrary function with an unlimited number of points [92]. A mathematical and statistical method is used to evaluate the neural network's performance. According to Equation (3), *MAE* were used to measure the accuracy of neural network results in this study to determine the network's performance.

$$MAE = \frac{1}{N} \sum_{i=1}^{N} \left| I_i - \hat{I}_i \right| \tag{3}$$

where $N$ represents the number of observations, $I_i$ represents the measured values and $\hat{I}_i$ represents the predicted values. The lower values of *MAE* indicate an accurate and optimal model.

### 3.2.4. Network Training and Hidden Layers

Learning is one of the most important capabilities of ANNs. There are two types of free parameters in each neural network: weights and errors. These parameters can be modified. It is imperative to determine the correct value of these parameters in order to ensure the network performs optimally in solving the problem. Consequently, the training of ANN-MLP is based on the concept that the free parameters are optimized by the training algorithms and based on the training data in such a way that the error value between the network output and the target parameter reaches the lowest possible value [93]. In general, there are two types of neural network training, supervised training, and unsupervised training. When training neural networks with a supervisor, the weights of neural networks are usually determined by defining a cost function and training on a set of experimental data referred to as training data [94]. The weights of a network in supervised training are determined in such a way that the cost functions are minimized. A method based on the trial and error rule known as error back propagation learning, which is used to train multi-layer perceptron networks, is a supervised training method. Multi-layer neural networks are commonly trained using the error back propagation learning algorithm [95].

The mechanism of error back propagation includes two main paths, the forward path and the backward path. On the forward path, a training pattern is applied to the network, followed by intermediate hidden layers, which propagate the training pattern to the output layer. As a result, the network parameters, including the residual error vectors and weights, are considered fixed until the actual output of the scheme is calculated. On the backward path, however, the network parameters are changed and reset. These changes are made based on trial and error [96]. An error vector represents the difference between the network's actual response and the desired response. As the error value is distributed throughout the entire network, from the output layer to other layers, the network weights are adjusted in such a way that the *MAE* of the network is minimized [97]. The error back propagation algorithm continues until, firstly, the *MAE* in each cycle is less than the predetermined value. This is because the amount of change in the network parameters after each cycle is extremely small, and secondly, the error gradient should be smaller than a predetermined value. As part of the error back propagation algorithm, three learning methods are available: a network with a simple learning coefficient, a network with a variable learning coefficient, and a network utilizing the Levenberg–Marquardt method [98]. A multi-layer perceptron neural network was employed using the Levenberg–Marquardt method in this study. It is important to note that entering training data in raw form reduces the speed and accuracy of the neural network. Therefore, to facilitate convergence, training data were standardized between zero and one using Equation (4):

$$v_{is} = \frac{x_i - x_{min}}{x_{max} - x_{min}} \tag{4}$$

where $v_{is}$ is represents the standardized value of the training data, $x_i$ is the input value of the training data, $x_{min}$ is the minimum value of the training data and $x_{max}$ is the maximum value of the training data.

It is crucial to prevent the network from overtraining in ANNs (overfitting problem). The network, in this case, maintains the relationships between the parameters instead of learning the relationships between them in order to generate a response in the output layer [99]. As part of achieving this goal, sampling data are categorized into three categories. The first category relates to training the network and generating an output layer response. As a second category of data, test data are used in order to control the amount of network error during the learning process. This is performed in order to determine if the neural network has been overtrained or not. Validation data (third category) are also used to determine whether the neural network has been approved after the learning process has been implemented.

In order to implement a perceptron neural network, it is imperative to determine the number of hidden layers after selecting the input layers and specifying the training

data. More hidden layers could improve the system's ability to understand the complexity of the problem. On the other hand, increasing the number of layers may reduce the prediction accuracy of the system and hinder its convergence. It is also important to take into consideration the number of neurons in each layer of the network. The neurons in the intermediate layers act as pattern recognizers [100]. In this way, the number of neurons at the hidden layer has a significant influence on the strength of the network. A low number of neurons reduces the power of analysis, reduces the numerical accuracy of prediction, and the network is not capable of establishing an accurate non-linear mapping between inputs and outputs. However, the excessive increase in neurons in the middle layers has resulted in a non-linear and complex mapping. This preserves the training data instead of analyzing them in this situation, which results in the network not performing optimally on new data as a result. The generalization power of the network decreases [101]. To resolve this problem, the number of neurons should be selected in such a way that the network has sufficient power but not too much. Thus, inputs will be mapped to outputs, and the network will operate at an optimal level.

## 4. Results

To implement ANN-MLP, the first step is to determine the input data and target. In this regard, the criteria of elevation, slope, distance from the fault, material type, skeleton type, distance from the fuel station, the density of vulnerable population, density of total population, distance from transmission lines, distance from the fire station, distance from the hospital, distance from the pharmacy, density of literate population, the density of working population, distance from the main road, and distance from public transportation station were inputs into the model and function of predicting vulnerable areas. The criteria were analyzed using spatial analysis to create a map of specific criteria. The criteria were divided into three groups: exposure (Figure 5), sensitivity (Figure 6), and adaptability capacity (Figure 7). The study was conducted at the level of urban blocks. The average values of each block were applied to distance criteria such as distance from medical centers, as well as height and slope criteria. This was performed to equalize the study level across urban blocks.

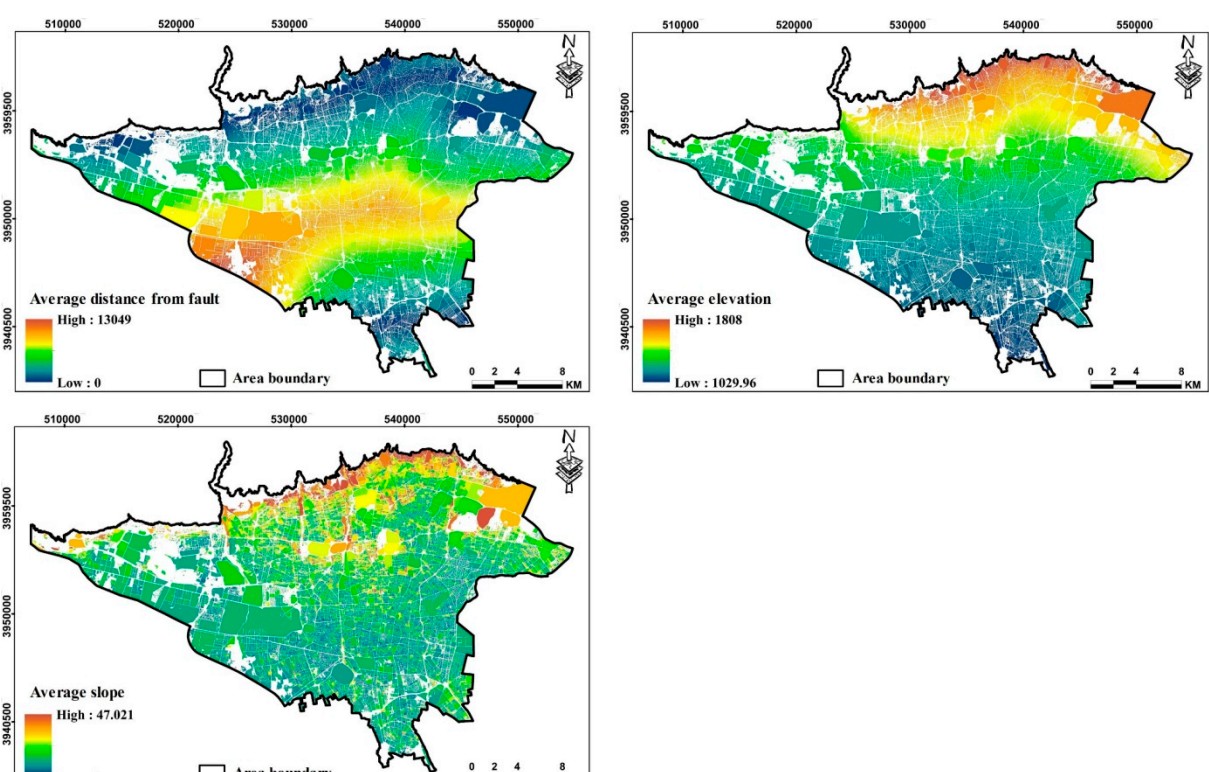

**Figure 5.** Map of exposure criteria.

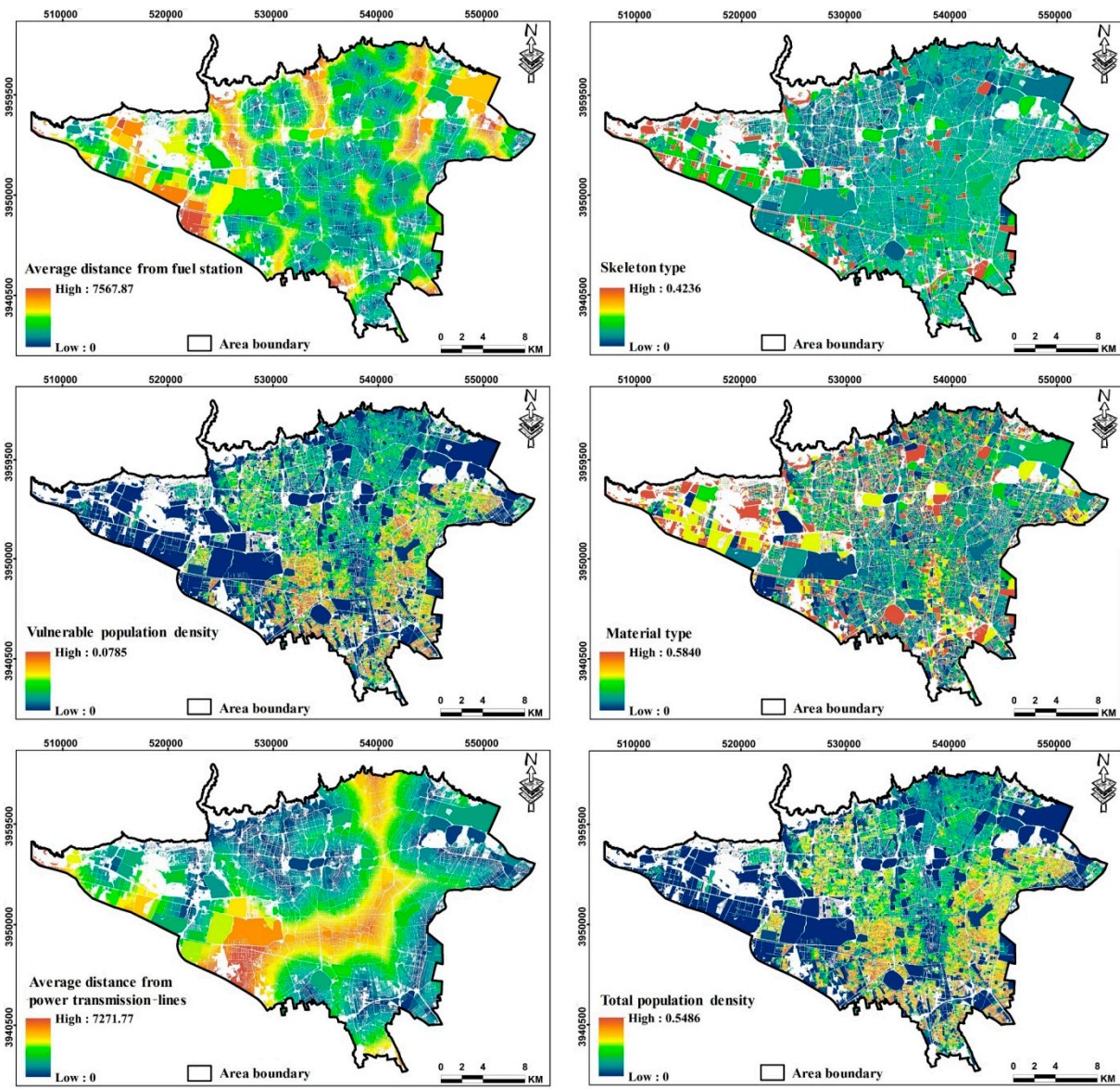

**Figure 6.** Map of sensitivity criteria.

After the set of effective criteria was selected to prepare the earthquake vulnerability map, each criterion was stored in the spatial database as a GIS map. The layers should be converted into comparable units in order to improve the training capability of the network and better understand the relationships between the input parameters [102]. As a result, the input parameters must be standardized to a numerical scale between zero and one before entering the neural network (zero values indicate very low vulnerability, and one values indicate very high vulnerability). To achieve this, all input parameters were standardized to a scale between zero and one using fuzzy functions, as shown in Figures 8–10. Fuzzification of each parameter was performed according to its purpose. As part of this process, elevation; slope; material type; skeleton type; the density of vulnerable population, density of total population; and distances from the fire station, medical center, pharmacy, main road, and public transportation station were standardized using the MSLarge function, since increasing the value of these variables has a greater impact on vulnerability. In contrast, the low values of distance from the fault, distance from the fuel station, distance from the power transmission line, literacy density, and working density have a greater impact on earthquake vulnerability. Due to this, the MSSmall function has been used for this category of criteria.

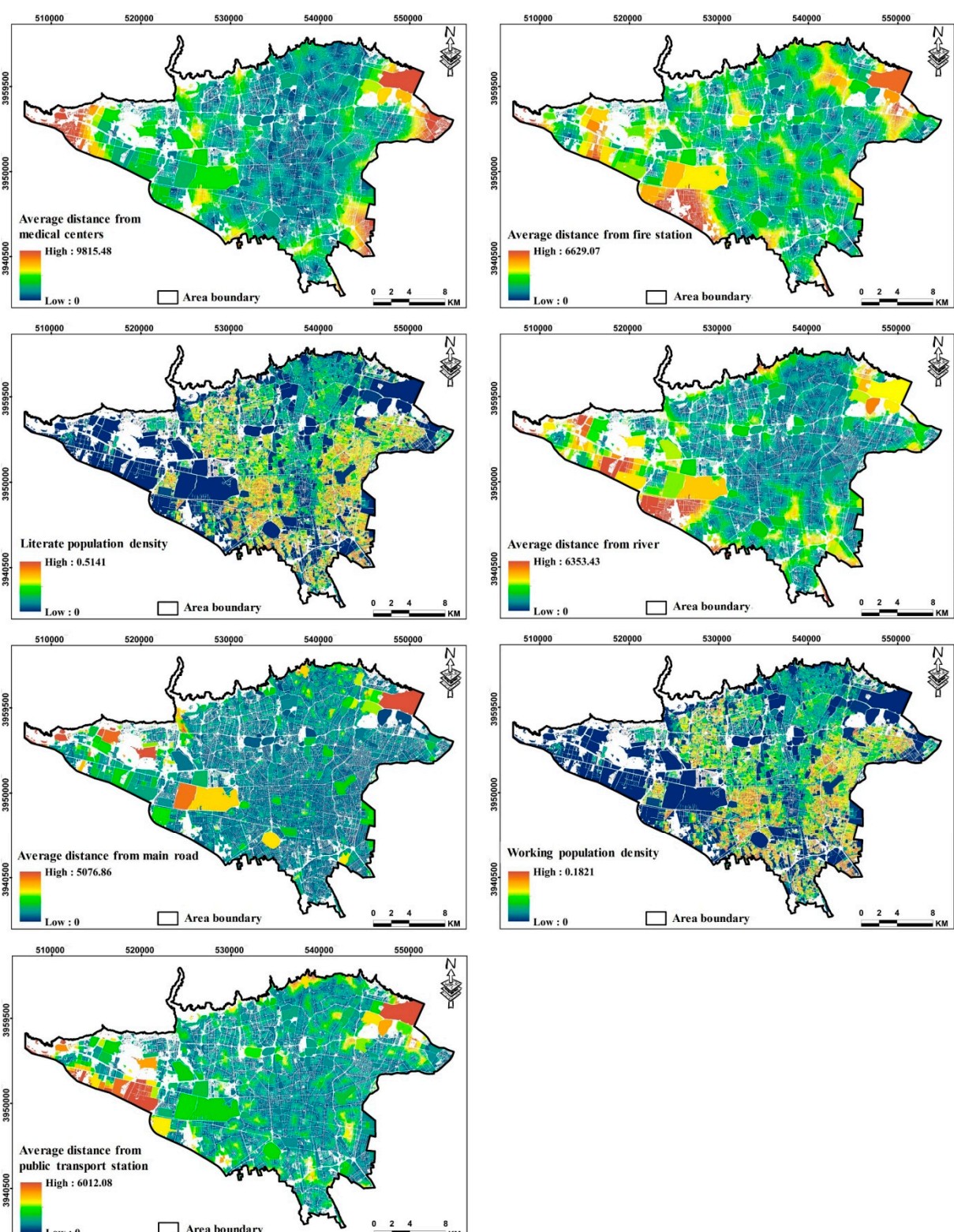

**Figure 7.** Map of Adaptability capacity criteria.

In order to model with ANNs, real data are used for training. During the modeling process, other unknown samples are compared with the training samples. Due to the lack of accurate earthquake statistics in the study area, a hybrid ANP-WLC process has been used to create a database of training data in this area. Consequently, the weights of each of the criteria were determined based on the opinions of experts (including urban management

experts, GIS experts, and earthquake engineering experts) and the ANP method (Table 3). Criteria weight values range from zero to one. Zero indicates the least importance, and one indicates the greatest importance. The sum of the values of all criteria equals one. Among the criteria selected to prepare an earthquake vulnerability map, vulnerable population density, total population density, and distance from fault have the highest weight and importance, while slope, elevation, and distance from public transport stations have the least weight and importance. Based on the obtained results, a consistency rate of less than 0.1 was determined for the calculation of the weight of criteria based on expert opinions. It basically demonstrates the compatibility and acceptance of the opinions of experts.

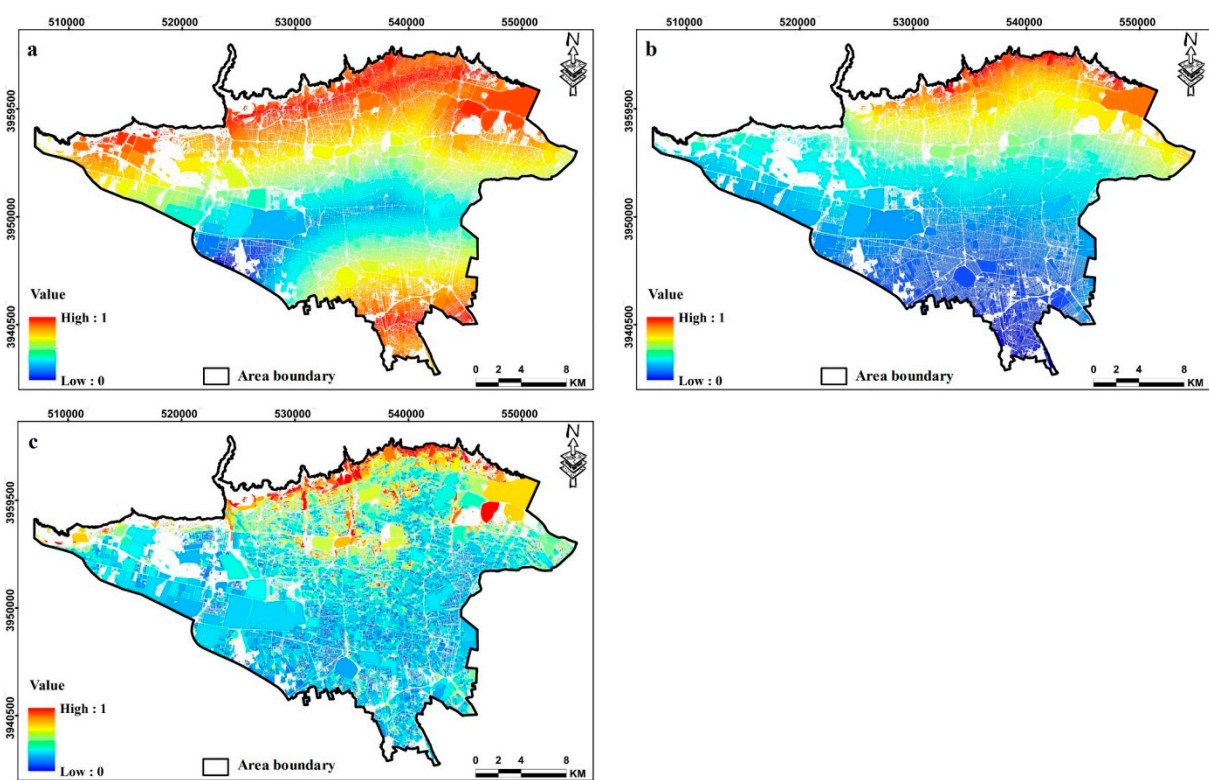

**Figure 8.** Fuzzification map of exposure criteria: (**a**) average distance from fault; (**b**) average elevation; and (**c**) Average slope.

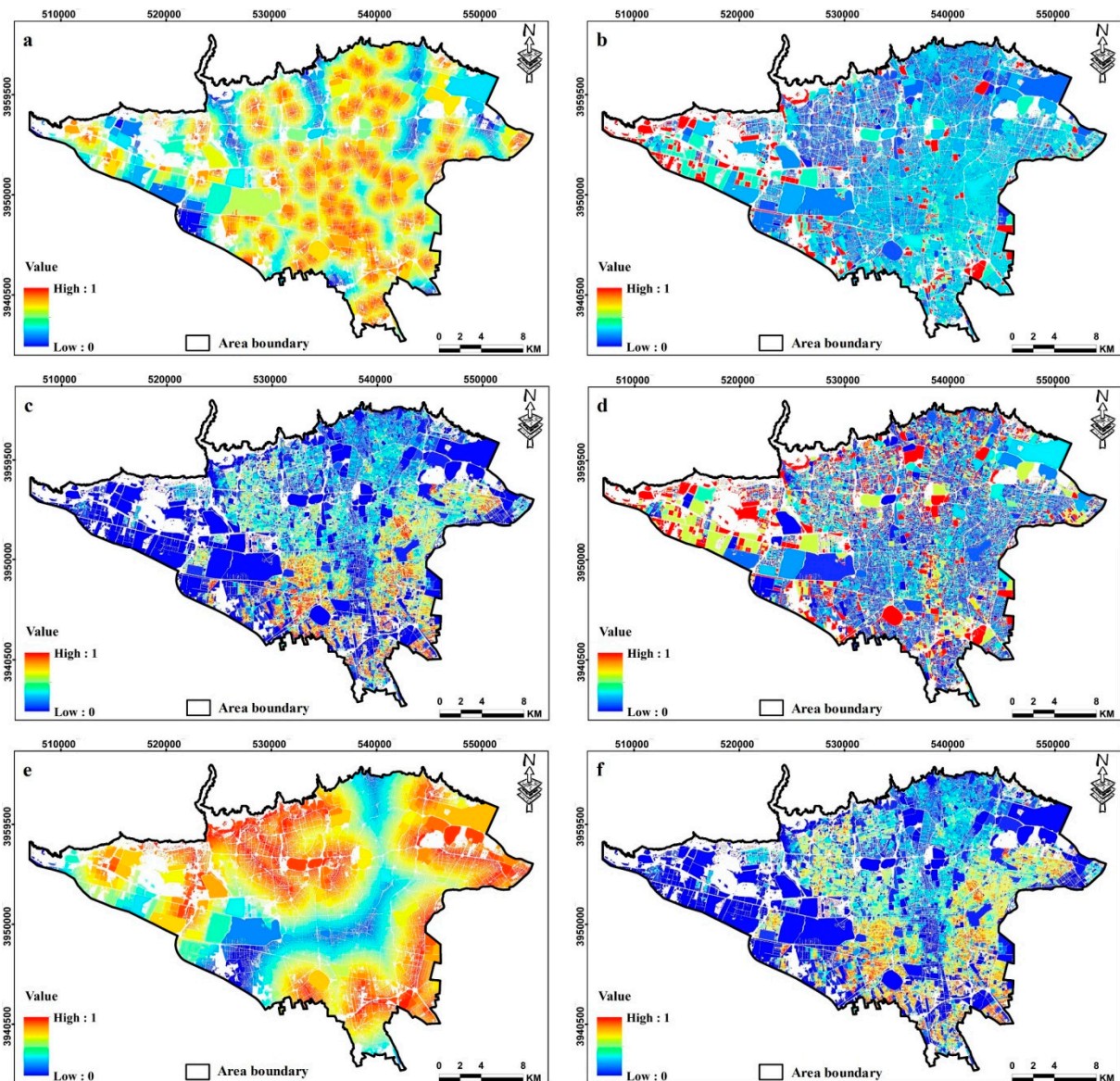

**Figure 9.** Fuzzification map of sensitivity criteria: (**a**) average distance from fuel station; (**b**) skeleton type; (**c**) vulnerable population density; (**d**) material type; (**e**) average distance from power transmission lines; (**f**) total population density.

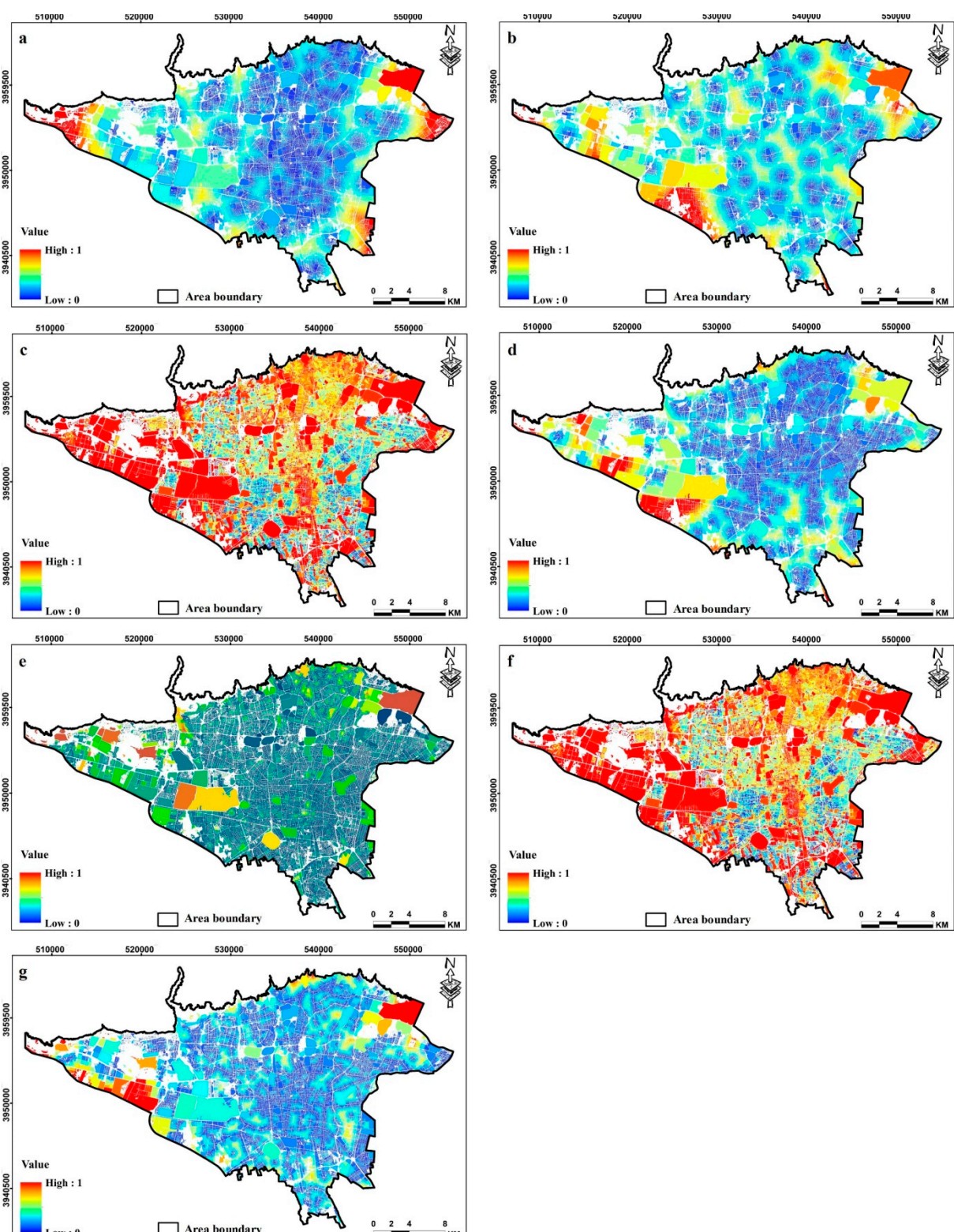

**Figure 10.** Fuzzification map of adaptability capacity criteria: (**a**) average distance from medical centers; (**b**) average distance from fire station; (**c**) literate population density; (**d**) average distance from river; (**e**) average distance from main road; (**f**) working population density; (**g**) average distance from public transport station.

**Table 3.** Weight and fuzzy function for the used criteria.

| Component | Criterion | Weight | Fuzzy Function | Type |
|---|---|---|---|---|
| Adaptability capacity | Distance from fire station | 0.03856 | MSLarge | Maximize |
| | Distance from medical centers | 0.07338 | MSLarge | Maximize |
| | Distance from pharmacy | 0.05923 | MSLarge | Maximize |
| | Density of literate population | 0.03685 | MSSmall | Minimize |
| | Working population density | 0.03255 | MSSmall | Minimize |
| | Distance from main road | 0.04369 | MSLarge | Maximize |
| | Distance from public transport station | 0.02992 | MSLarge | Maximize |
| Exposure | Elevation | 0.02555 | MSLarge | Maximize |
| | Distance from fault | 0.10356 | MSSmall | Minimize |
| | Slope | 0.02136 | MSLarge | Maximize |
| Sensitivity | Skeleton type | 0.09852 | MSLarge | Maximize |
| | Material type | 0.09234 | MSLarge | Maximize |
| | Distance from fuel station | 0.04736 | MSSmall | Minimize |
| | Vulnerable population density | 0.12932 | MSLarge | Maximize |
| | Total population density | 0.11653 | MSLarge | Maximize |
| | Distance from power transmission lines | 0.05128 | MSSmall | Minimize |

We generated an earthquake vulnerability map for Tehran using the standardized maps and weights of criteria as input to the WLC model (Figure 11). A correlation of each raster layer with the input parameters of the model with respect to the output layer was calculated separately in order to verify the output of the combined ANP-WLC model and to ensure that the generated training database was accurate. Positive numbers indicate a direct correlation between the two analyzed parameters, while negative numbers indicate an indirect and inverse correlation. A greater absolute value of the correlation coefficient indicates a greater correlation between two parameters. It can be observed from Table 4 that the criteria that have the greatest weight and impact on vulnerability in ANP have a higher correlation with the ANP-WLC hybrid model output. We may then conclude that the combined map of the ANP-WLC model can be applied to creating the training database with reasonable accuracy. Through the implementation of a hybrid ANP-WLC model and its validation using the correlation method, a database of training data for ANNs was created using the output map. A total of 167 training data were collected and inserted in the model, of which 70% (117 points) were used for training, 15% (25 points) for model testing, and the remaining 15% (25 points) for model validation. All extracted points were transferred to Google Earth, and their coordinates were determined. After referring to 15 locations and conducting a field investigation, it was discovered that they had already been damaged by previous earthquakes.

In this study, an ANN-MLP with 16 input layers (underlying factors in earthquake vulnerability), 10 hidden layers (trial and error method), and 1 neuron in the output layer (vulnerability map) was built. The network was then trained using Levenberg–Marquardt learning. The network was stopped after 25 iterations and achieved a sufficient level of training. Moreover, the network had reached an optimal state in the 10th iteration, characterized by the highest correlation and lowest error. Furthermore, the *MAE* value was equal to 0.085. Figure 12 shows the fitting diagram and regression coefficient for the training stages, validation, and final testing of the neural network. As a result, optimal numerical values were obtained for these steps.

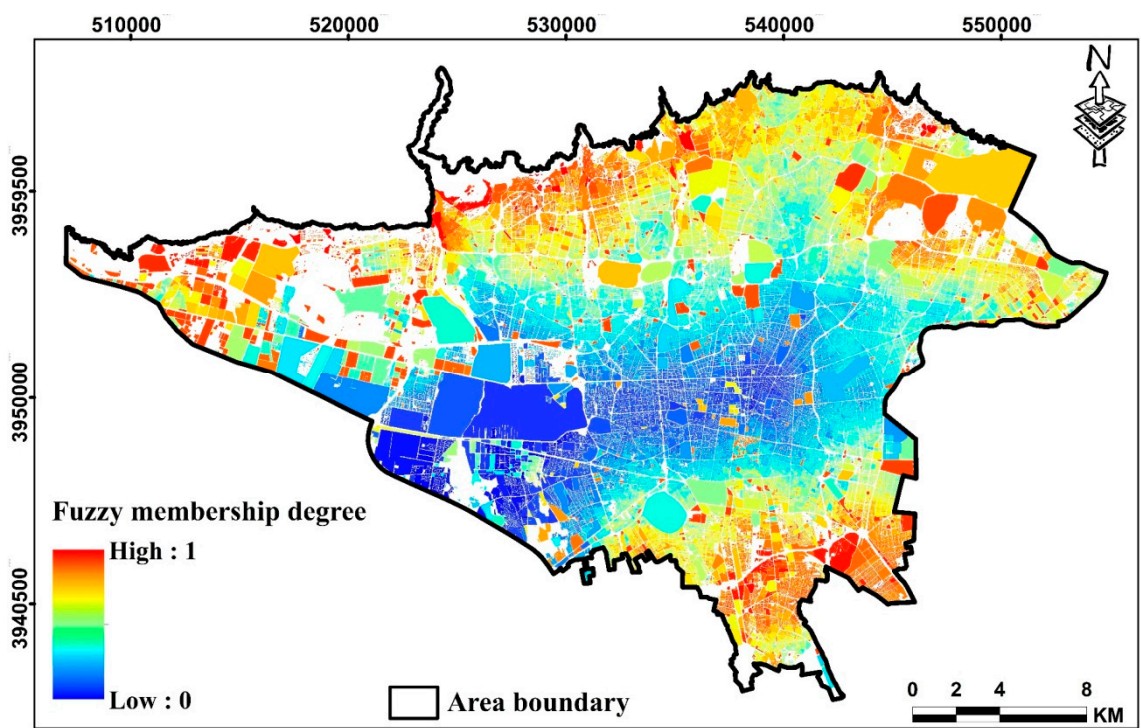

**Figure 11.** Earthquake vulnerability map in urban blocks using the ANP-WLC model. Higher degrees of fuzzy membership in this map indicate higher vulnerability.

**Table 4.** Correlation between the criteria and the output of the ANP-WLC model.

| Parameter | Correlation Coefficient | Parameter | Correlation Coefficient |
|---|---|---|---|
| Distance from fire station | 0.55 | Distance from fault | −0.95 |
| Distance from medical centers | 0.65 | Slope | 0.46 |
| Distance from pharmacy | 0.63 | Skeleton type | 0.82 |
| Literate population density | −0.66 | Material type | 0.79 |
| Working population density | −0.69 | Distance from fuel station | −0.57 |
| Distance from main road | 0.50 | Vulnerable population density | 0.77 |
| Distance from public transport station | 0.53 | Total population density | 0.75 |
| Elevation | 0.49 | Distance from power transmission lines | −0.75 |

Based on the correlation coefficient (0.90) obtained for the validation step in Figure 12, it can be concluded that the network has achieved a satisfying level of learning. Finally, the overall correlation coefficient of the network, which is calculated based on the inclusion of all data in the network, was estimated to be 0.92. Following the completion of the learning process, the network was able to value new regions in accordance with what it had learned. Following the completion of the learning process, the entire study area was provided to the trained network. In order to determine vulnerable areas, the network evaluated the entire study area based on the weights obtained from the input parameters during the learning phase. According to Figure 13, this step produces a fuzzy map with variable degrees of fuzzy membership ranging from zero to one. Thus, a higher degree of membership indicates higher vulnerability when dealing with earthquakes.

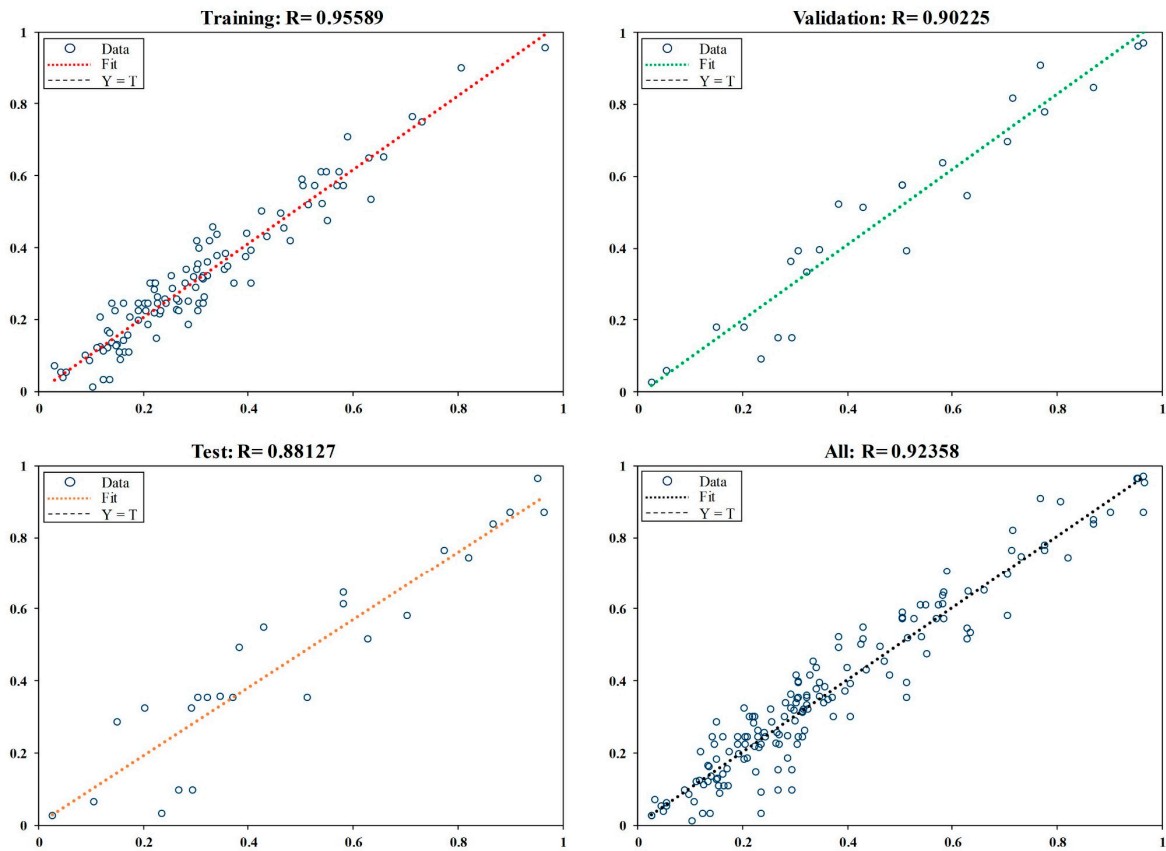

**Figure 12.** The obtained fitting diagrams and the related correlation coefficients in different stages of the network.

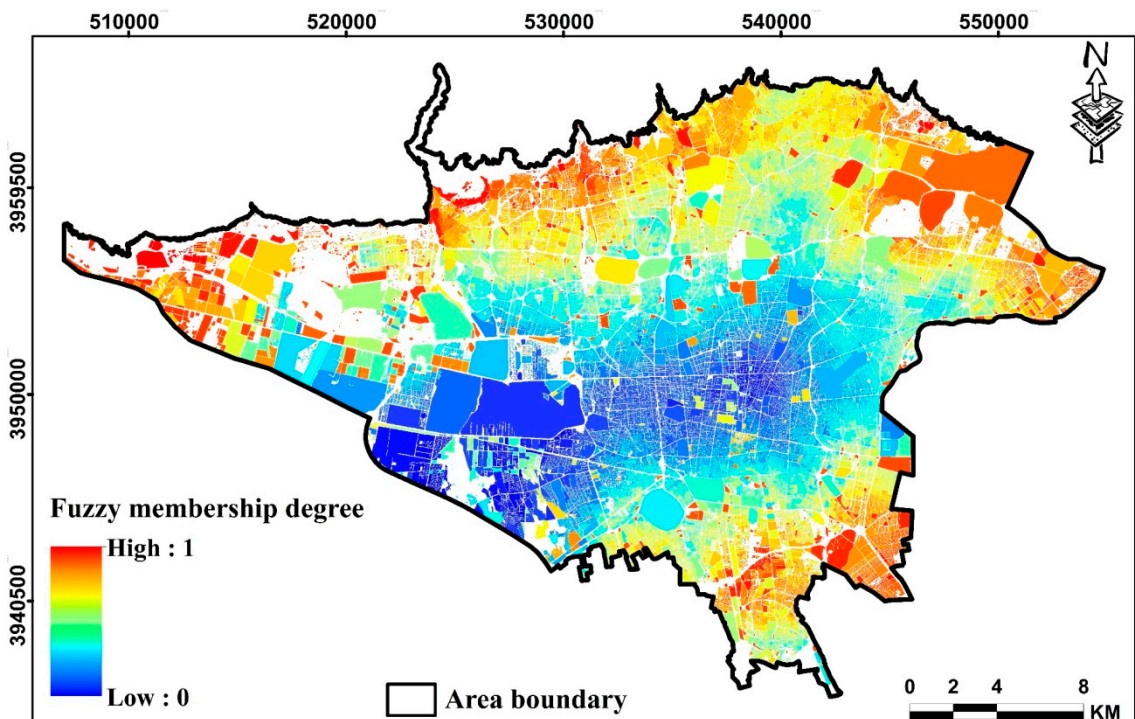

**Figure 13.** Earthquake vulnerability map over city blocks using ANN-MLP model. Higher degrees of fuzzy membership in this map indicate higher vulnerability.

Based on the output of an ANN-MLP model, Figure 14 illustrates the spatial distribution of earthquake-vulnerable areas. There were five different vulnerability classes in the study area: very high vulnerability (0–0.2), high vulnerability (0.2–0.4), medium vulnerability (0.4–0.6), low vulnerability (0.6–0.8), and very low vulnerability (0.8–1). Figure 14 indicates that vulnerable areas of the study area are deliberately located in the northern and southern regions. Additionally, the northern regions have high elevations and steep slopes, which can contribute significantly to the region's vulnerability. Additionally, they are most close to fault lines and have the most vulnerable populations. According to the area of different vulnerability classes, the largest area is in the medium vulnerability class (32% of the total area of the study area). Only 5% of the total area of the study area is classified as very high vulnerability. Based on the vulnerability map, a large area of Tehran is in the medium to very high vulnerability class; therefore, mitigation measures should be taken into account.

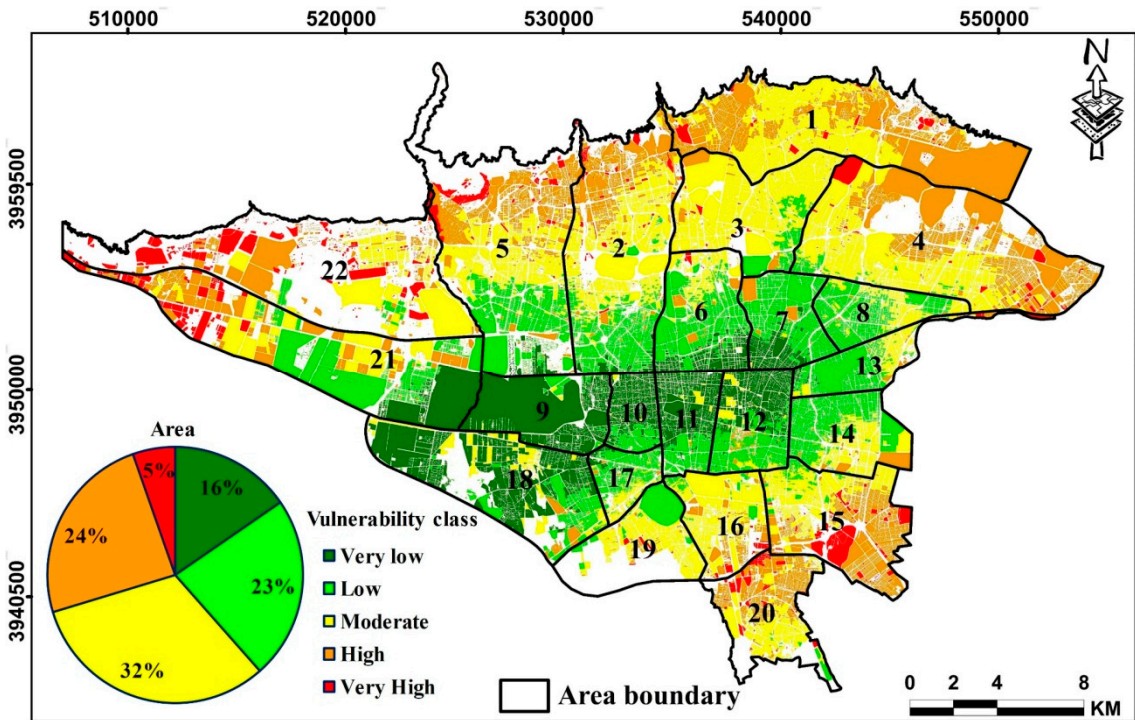

**Figure 14.** Classified map of spatial distribution and percentage of vulnerable areas to earthquakes (numbers 1 to 22 represent the urban districts of Tehran).

For the 22 districts of Tehran, the command Zonal Statistics as Table has been used to obtain the area of various classes. Based on the results shown in Table 5, districts 9, 8, 3, 1, and 22 have the highest number of very low, low, medium, high, and very high vulnerable classes, respectively. Therefore, 94% of the area of District 9 is in the very low class, 73% of the area of District 8 is in the low class, 80% of the area of District 3 is in the medium class, 60% of the area of District 1 is in high class, and 27% of the area of District 22 is in very high class.

**Table 5.** The percentage of the area of different vulnerable classes for the 22 districts of Tehran.

| | Vulnerability Class | | | | |
|---|---|---|---|---|---|
| **Districts** | **Very Low** | **Low** | **Moderate** | **High** | **Very High** |
| 1 | 0 | 0 | 37 | 60 | 3 |
| 2 | 2 | 24 | 44 | 26 | 3 |
| 3 | 0 | 13 | 80 | 6 | 1 |

**Table 5.** *Cont.*

| Districts | Vulnerability Class | | | | |
|---|---|---|---|---|---|
| | **Very Low** | **Low** | **Moderate** | **High** | **Very High** |
| 4 | 0 | 3 | 43 | 49 | 5 |
| 5 | 3 | 30 | 29 | 31 | 7 |
| 6 | 19 | 59 | 16 | 5 | 1 |
| 7 | 23 | 68 | 1 | 9 | 0 |
| 8 | 0 | 73 | 26 | 1 | 0 |
| 9 | 94 | 5 | 1 | 0 | 0 |
| 10 | 70 | 28 | 2 | 0 | 0 |
| 11 | 68 | 29 | 3 | 0 | 0 |
| 12 | 46 | 42 | 9 | 3 | 0 |
| 13 | 3 | 67 | 22 | 4 | 3 |
| 14 | 0 | 60 | 29 | 11 | 0 |
| 15 | 0 | 6 | 33 | 45 | 17 |
| 16 | 0 | 7 | 71 | 18 | 4 |
| 17 | 9 | 67 | 22 | 3 | 0 |
| 18 | 66 | 17 | 13 | 4 | 0 |
| 19 | 0 | 38 | 56 | 4 | 2 |
| 20 | 0 | 3 | 41 | 45 | 11 |
| 21 | 19 | 32 | 19 | 22 | 9 |
| 22 | 0 | 4 | 42 | 28 | 27 |

## 5. Discussion

The vulnerability of individuals and infrastructure to earthquakes, particularly in dense urban areas, has increased due to intense urbanization and the increase in high-rise buildings [103]. In order to take alarming and preventive measures and mitigate damages caused by earthquakes, it is a vital task to map out vulnerable areas and take precautionary measures to minimize the risks, which is called vulnerability assessment in the disaster management cycle [104].

In this study, several prominent criteria were examined in three categories, including exposure (three criteria), sensitivity (six criteria), and adaptability capacity (seven criteria). First, by using the ANP-WLC combination, an earthquake vulnerability map was produced for the studied area. Based on the correlation obtained between the criteria map and the generated vulnerability map, as well as the field test of the validation model and the creation of a database of training points for the implementation of the ANN, a validation model was developed. The general conclusion from the present research indicates that the combined use of multi-criteria decision-making methods, ANN and GIS, can facilitate the identification of vulnerable areas well. This method can be used to identify vulnerable areas in other areas. The areas with high vulnerability in this study were also compared with the results of Hashemi et al. [105]. The vulnerability map produced by Hashemi et al. [105] covers the vulnerable areas in the present study. Moreover, the results of our study show reasonable agreement with the vulnerability map prepared by Moradi et al. [106].

Our findings clearly show that the northern areas of Tehran city are highly vulnerable to earthquakes. This is due to their proximity to the Mosha and North Tehran faults, leveraged by their elevation and steep slopes. Moreover, Districts 15 and 20, located in the south of Tehran, have a large area of high vulnerability due to their proximity to the Ray fault. Due to a lack of attention to the quality and durability of structural materials,

high-rise buildings are highly vulnerable, and most of them are categorized within the high vulnerability class. Moreover, open space plays an influential role in mitigating vulnerability because buildings with a higher area have less vulnerability. Districts 9–11 have low vulnerability due to sufficient distance from the faults and proper access to treatment and transportation facilities. Furthermore, 94% of district 9, 70% of district 10, and 68% of district 11 are in the class of "very low" vulnerability. The results of this part of our study are consistent with those reported by Kamranzad et al. [107] and Nazmfar et al. [108] but are not consistent with what was reported by Hajibabaee et al. [109] and Rezaie and Panahi [110]. This can be attributed to the difference in input data and the methods used. In the studies conducted by Hajibabaee et al. [109] and Rezaie and Panahi [110], the vulnerability related only to the occurrence of earthquakes on the Ray and Mosha faults, while we included a wider set of spatial data as well as all the faults around Tehran city. The overall design of the hybrid MCDA-ANN model developed in this study is scalable because it is an integrated approach to earthquake vulnerability assessment. The outcome of our study can be used by planners and managers and any other decision maker in the area of retrofitting buildings, optimizing access, determining suitable places for temporary accommodation post-earthquake, and establishing relief stations according to the vulnerability of areas.

In past studies, a number of criteria, such as earthquake-induced peak acceleration of ground displacement [111], street width [108], soil characteristics [112], and geology [113], have been considered to evaluate the vulnerability caused by earthquakes. However, in this study, due to limited access to related data, it was not possible to use these criteria in the vulnerability assessment process. Consequently, we suggest considering the impact of these criteria for preparing the vulnerability map in future studies. Since ANN is rather a black box and difficult to explain, we further recommend using more explainable models that model designers can cautiously modify.

## 6. Conclusions

Recognizing the vulnerability of urban areas to earthquakes, as they remain a significant natural hazard that can have devastating impacts, is of great importance across the world. The use of ANN-MLP and the combined ANP-WLC method in this study allowed for the creation of a detailed vulnerability map for Tehran, which can be used to inform efforts to reduce vulnerability and mitigate risks. These efforts are particularly important given that certain districts in Tehran, such as districts 1 and 22, are identified as having a higher proportion of vulnerable classes. It is also worth noticing that vulnerability to earthquakes is not static. This is due to a variety of factors, such as the age and condition of buildings, population density, and the availability of resources for hazard mitigation and emergency response. As such, it is important to continuously assess and update vulnerability maps to ensure that the necessary measures are in place to protect urban areas from earthquakes. In addition to mapping the vulnerability of urban blocks, it would also be beneficial to map vulnerable populations, infrastructure, and business sectors in order to prioritize the development of resilient buildings and emergency response efforts. This can help to ensure that the most vulnerable members of the community are protected during and after an earthquake. While substantial efforts have been made to strengthen urban buildings and increase the knowledge and abilities of specialists in managing natural hazards, earthquakes remain a serious threat to urban areas. The findings of our study highlight the importance of continued efforts to reduce vulnerability and mitigate risks in order to protect communities from the destructive impacts of earthquakes.

**Author Contributions:** R.A., supervision and writing—review and editing; S.N.S., data curation, formal analysis, investigation, methodology, resources, software, writing—original draft, and writing—review and editing; A.R.B.L., data collection and writing—review and editing; M.H. and J.J.A., writing—review and editing. All authors have read and agreed to the published version of the manuscript.

**Funding:** This research received no external funding.

**Data Availability Statement:** The data used to support the findings of this study are available from the corresponding author upon reasonable request.

**Acknowledgments:** This study was supported by the Agrohydrology Research Group of Tarbiat Modares University (Grant No. IG-39713).

**Conflicts of Interest:** The authors declare no conflict of interest.

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
