# Peer review of "Using Artificial Neural Networks to Assess Earthquake Vulnerability in Urban Blocks of Tehran"

_remotesensing, doi:10.3390/rs15051248_

Round 1

Reviewer 1 Report

 1.     The entire document needs a major revision in terms of structuring. There is a need to synthesize based on the literature. This introduction should include a brief description and a justified choice of the possible tools/analytical methods. There also is a need to revisit the methods, results, and discussion sections extensively.

2.     Please consider a thorough revision of the text, as there are many punctuation problems, missing words, vague formulations, etc.

3.     The overall conceptualization of the research needs considerable strengthening.

4.     The use of literature is limited; there is little justification for the choices made. The latest work on deep learning and other XAi-based models in geohazard modeling should be well justified.

5.     The selection of the study area and parameters used to need more justification.

6.     ANN-based models are considered black-box; how have authors addressed this issue?

7.     The authors should enhance the discussion section by talking about: (i) the implications of their findings in the context of the current trend (and necessity) and (ii) can the approach of the present paper has practical usage for planners and how their findings can be communicated. In other words, the authors should emphasize the added value of their paper in the current context.

8.     Reporting of results generated is overall correct, but the problem is a lack of interpretation of the outcomes. Reflection and synthesis should be strengthened.

Remarks: The study in its current form is not fit for publication and would need a very major revision to become publishable.

Author Response

Our response follows:

Reviewer #1:

Dear reviewer

We appreciate your time and effort in giving valuable comments on our paper. Indeed, the changes we made to this paper on the basis of your questions, have added to the scientific value of the paper. Following you see Responses to your questions in the same order of appearance.

Point 1:  The entire document needs a major revision in terms of structuring. There is a need to synthesize based on the literature. This introduction should include a brief description and a justified choice of the possible tools/analytical methods. There also is a need to revisit the methods, results, and discussion sections extensively.

Response:

First, we appreciate you very much for your feedback on our manuscript. Based on your comments, the entire manuscript has been revised. Also, explanations about the tools and methods used have been added in the introduction section (Please see: page 2, lines 61-91). A new section entitled "Discussion" is now added to the revised manuscript (Please see: pages 22 and 23, lines 505-549).

Point 2: Please consider a thorough revision of the text, as there are many punctuation problems, missing words, vague formulations, etc.

Response:

Thanks for this precise comment. The manuscript has been reviewed and revised by a native language editor.

Point 3: The overall conceptualization of the research needs considerable strengthening.

Response:

Thanks for this precise comment. The overall conceptualization of the research was strengthened (Please see: page 2, lines 49-61).

Point 4: The use of literature is limited; there is little justification for the choices made. The latest work on deep learning and other XAi-based models in geohazard modeling should be well justified.

Response:

Thank you for your suggestion. Based on your comment, some further relevant studies have been used in the literature (Please see: page 3, lines 117-133).

Point 5: The selection of the study area and parameters used to need more justification.

Response:

Thank you for this point.  In relation to the reason for choosing the study area and the parameters used, some information has been added in the revised manuscript (Please see: page 3, lines 142-147, and pages 7 and 8, Table 2).

Point 6: ANN-based models are considered black-box; how have authors addressed this issue?

Response:

Thanks for this precise comment. As mentioned by the respected reviewer, some machine learning methods are black box (for example, ANN). But it should be considered that the effectiveness of the neural network method has been proven in various studies by Prabu & Ramakrishnan, (2009), Quan & Lee, (2012), Yeo & Yee, (2014), Borgogno Mondino et al., (2015), Alizadeh et al., (2018), Dianati Tilaki et al., (2020) and Jena & Pradhan, (2020).

The results of our study showed that despite the black box limitation in the ANN model, this model has high efficiency in different modeling and can be more accurate than conventional methods. However, at the end of the discussion section, we have pointed out that in future studies, the effectiveness of black box methods need to be evaluated with non-black box methods (Please see: page 23, lines 548 and 549).

References

  1. Prabu, S., & Ramakrishnan, S. S. (2009). Combined use of socio economic analysis, remote sensing and GIS data for landslide hazard mapping using ANN. Journal of the Indian Society of Remote Sensing37, 409-421.
  2. Quan, H. C., & Lee, B. G. (2012). GIS-based landslide susceptibility mapping using analytic hierarchy process and artificial neural network in Jeju (Korea). KSCE Journal of Civil Engineering16, 1258-1266.
  3. Yeo, I. A., & Yee, J. J. (2014). A proposal for a site location planning model of environmentally friendly urban energy supply plants using an environment and energy geographical information system (E-GIS) database (DB) and an artificial neural network (ANN). Applied Energy119, 99-117.
  4. Borgogno Mondino, E., Fabrizio, E., & Chiabrando, R. (2015). Site selection of large ground-mounted photovoltaic plants: a GIS decision support system and an application to Italy. International Journal of Green Energy12(5), 515-525.
  5. Alizadeh, M., Ngah, I., Hashim, M., Pradhan, B., & Pour, A. B. (2018). A hybrid analytic network process and artificial neural network (ANP-ANN) model for urban earthquake vulnerability assessment. Remote Sensing10(6), 975.
  6. Dianati Tilaki, G. A., Ahmadi Jolandan, M., & Gholami, V. (2020). Rangelands production modeling using an artificial neural network (ANN) and geographic information system (GIS) in Baladeh rangelands, North Iran. Caspian Journal of Environmental Sciences18(3), 277-290.
  7. Jena, R., & Pradhan, B. (2020). Integrated ANN-cross-validation and AHP-TOPSIS model to improve earthquake risk assessment. International Journal of Disaster Risk Reduction50, 101723.

Point 7: The authors should enhance the discussion section by talking about: (i) the implications of their findings in the context of the current trend (and necessity) and (ii) can the approach of the present paper has practical usage for planners and how their findings can be communicated. In other words, the authors should emphasize the added value of their paper in the current context.

Response:

Thanks again for this precise comment. Based on your comment, a new section entitled "Discussion" was added to the revised manuscript, and your comments were addressed in this section (Please see: pages 22 and 23, lines 505-549).

Point 8:  Reporting of results generated is overall correct, but the problem is a lack of interpretation of the outcomes. Reflection and synthesis should be strengthened.

Response:

Thanks for this precise comment. As much as possible, the results section was strengthened.

Reviewer 2 Report

The paper uses  artificial neural networks to Assess Earthquake Vulnerability in Urban Blocks of Tehran.  It considers the following factors: Exposure, Sensitivity, Adaptive capacity. The integrating approach used in the paper is interesting. Yet, the following are needed in order to improve the quality of the paper:

(1) The manner with which the weight of each criterion must be better illustrated.

(2) The manner that validation , or training, of the proposed model is performed is not clear and must be better illustrated.

 (3) Regarding the exposure, (a) it is not clear which physical quantity is considered, such as earthquake-induced peak acceleration of ground displacement, (b) it is not clear how elevation affects exposure and (c) soil amplification, which depends on the soil profile of the area is a critical factor which must be considered for better predictions. The above, must be explained, at least in a discussion section, with the relevant references.

REFERENCES

Katsenis LC, CA Stamatopoulos, VP Panoskaltsis, B Di. (2020). Prediction of large seismic sliding movement of slopes using a 2-body non-linear dynamic model with a rotating stick-slip element. Soil Dynamics and Earthquake Engineering 129, 105953

Li B, Cai Z, Xie W, Pandey M (2018) Probabilistic seismic hazard analysis considering site-specific soil effects. Soil Dynamics and Earthquake Engineering 105:103–113, DOI Pehlivan

M, Rathje EM, Gilbert RB (2016) Factors influencing soil surface seismic hazard curves. Soil Dynamics and Earthquake Engineering 83:180–190

Author Response

Our response follows:

Reviewer #2:

Dear reviewer

We appreciate your time and effort in giving valuable comments on our paper. Indeed, the changes we made to this paper on the basis of your questions, have added to the scientific value of the paper. Following you see Responses to your questions in the same order of appearance.

The paper uses artificial neural networks to Assess Earthquake Vulnerability in Urban Blocks of Tehran.  It considers the following factors: Exposure, Sensitivity, and Adaptive capacity. The integrating approach used in the paper is interesting. Yet, the following are needed in order to improve the quality of the paper:

Point 1: The manner with which the weight of each criterion must be better illustrated.

Response:

First, we appreciate you very much for your feedback on our manuscript.

Corrected (Please see: page 18, Table 3).

Point 2: The manner that validation, or training, of the proposed model is performed is not clear and must be better illustrated.

Response:

Thanks for this precise comment. Based on the reviewer's comment, details related to the number of data used for training, testing and validation of the ANN model have been modified in the manuscript (Please see: pages 10 and 11, lines 317-352 and pages 18 and 19, lines 449-457). Also, the results related to the correlation coefficient of ANN model and training, testing and validation data are shown in Figure 12.

Point 3: Regarding the exposure, (a) it is not clear which physical quantity is considered, such as earthquake-induced peak acceleration of ground displacement, (b) it is not clear how elevation affects exposure and (c) soil amplification, which depends on the soil profile of the area is a critical factor which must be considered for better predictions. The above, must be explained, at least in a discussion section, with the relevant references.

References

Katsenis LC, CA Stamatopoulos, VP Panoskaltsis, B Di. (2020). Prediction of large seismic sliding movement of slopes using a 2-body non-linear dynamic model with a rotating stick-slip element. Soil Dynamics and Earthquake Engineering 129, 105953

Li B, Cai Z, Xie W, Pandey M (2018) Probabilistic seismic hazard analysis considering site-specific soil effects. Soil Dynamics and Earthquake Engineering 105:103–113, DOI Pehlivan

M, Rathje EM, Gilbert RB (2016) Factors influencing soil surface seismic hazard curves. Soil Dynamics and Earthquake Engineering 83:180–190

Response:

Many thanks for this precise comment and suggested references. According to your opinion, new information about effective and usable criteria in the field of earthquake vulnerability assessment was added to the discussion section, which was not considered in this study due to the limitation of access to this data. Also, your suggested references have been used in the revised manuscript (Please see: page 23, lines 543-547).

Reviewer 3 Report

I have several concerns about the current framework as follows:

1- It is not clear to me, what is the role of fuzzy logic (why fuzzy is used for normalization). And what are the grid values based in Figure 5? What exactly is the input for the neural network? 

2- For Eqs. 1-3, the authors did not use the activation function after each layer? Also, what are the number of input features, hidden layers, and neurons per layer? At which criteria did the authors choose the current neural network architecture? In other words, what will be the performance when the number of hidden layers and neurons changes?

3- Eq.1, I did not get what method was used to optimize the network parameters? SGD, Adam, or another optimization method?

4- Eq.2, looks weird! What R denotes here? And is there only one bias (b) for all the networks?

5- The authors stated “such a way that the RSME of the network is minimized”, however EQ. 3 shows the loss function used in this study is MAE!

6- I do not understand the roles of the neural network here. The authors already obtained the vulnerability map using the fuzzy algorithm as shown in Figure 12. Then they train the neural using the obtained vulnerability map! The sequence is confusing! In this way what are the advantages will the neural network will add? The highest performance of the neural network will be limited by the optimal solution obtained by the fuzzy algorithm.

7- The authors used 25 points for the test. What is the distribution of those points on the map? They are located in different areas or they are near each other?

8- The neural network has a point prediction, how do the authors obtain the vulnerability values for the rest of the map? Has any type of interpolation been done?

9- Is there any reference to the vulnerability map of Tehran to compare the results reached by the authors and interpret their results? 

Overall the paper is interesting and needs major revision!

Best,

Author Response

Our response follows:

Reviewer #3:

Dear reviewer

We appreciate your time and effort in giving valuable comments on our paper. Indeed, the changes we made to this paper on the basis of your questions, have added to the scientific value of the paper. Following you see Responses to your questions in the same order of appearance.

Point 1: It is not clear to me, what is the role of fuzzy logic (why fuzzy is used for normalization). And what are the grid values based in Figure 5? What exactly is the input for the neural network?

Response:

First of all, we have to be fully thankful for your positive review of our review work.  Because the scale and unit of the various criteria used in this study to prepare the earthquake vulnerability map are different, in order to use these criteria as inputs to the MCDA and ANN models, all the criteria must be standardized in a single scale. Therefore, in this study, fuzzy logic has been used to standardize the values of different criteria.

Also, Figure 5 (in the revised version of Figure 4) has been updated. The input of the neural network is the Fuzzification criteria and training data extracted from the MCDA method (Please see: page 10, Figure 4).

Point 2: For Eqs. 1-3, the authors did not use the activation function after each layer? Also, what are the number of input features, hidden layers, and neurons per layer? At which criteria did the authors choose the current neural network architecture? In other words, what will be the performance when the number of hidden layers and neurons changes?

Response:

Thanks for this precise comment. The MAE between the ANN output and the training data extracted from the MCDA method has been used as the objective function. A low value of MAE indicates the optimality of the ANN architecture (Please see: page 11, lines 331-343 and page 19, lines 462-644).

Point 3: Eq.1, I did not get what method was used to optimize the network parameters? SGD, Adam, or another optimization method?

Response:

Many thanks for this comment. Levenberg – Marquardt method is used to determine the optimal values of unknown parameters in ANN architecture, including the number of neurons, hidden layers, weights of criteria, etc. (Please see: page 11, lines 343-346 and page 19, lines 464-467).

Point 4: Eq.2, looks weird! What R denotes here? And is there only one bias (b) for all the networks?

Response:

Thanks for this nice comment. Based on your comments, the equation structure has been modified. A definition of its components was also provided.  is the bias of neuron  (Please see: page 9, lines 294-302).

Point 5: The authors stated “such a way that the RSME of the network is minimized”, however EQ. 3 shows the loss function used in this study is MAE!

Response:

Corrected (Please see: page 11, lines 339 and 340).

Point 6: I do not understand the roles of the neural network here. The authors already obtained the vulnerability map using the fuzzy algorithm as shown in Figure 12. Then they train the neural using the obtained vulnerability map! The sequence is confusing! In this way what are the advantages will the neural network will add? The highest performance of the neural network will be limited by the optimal solution obtained by the fuzzy algorithm.

Response:

Thanks for this comment. The neural network model has higher generalizability than the MCDA model. The developed model can be used in different areas. To implement this model, only the geographical location of a number of actual data damaged by the earthquake is needed to train the model and build its optimal architecture. In this case, the MCDA model has a high dependence on the weights determined by experts and has low generalizability. In this study, MCDA was used to generate training data due to the lack of access to the appropriate number of real earthquake vulnerability data. In other areas, it is not necessary to implement the MCDA model.

Point 7: The authors used 25 points for the test. What is the distribution of those points on the map? They are located in different areas or they are near each other?

Response:

Many thanks for this precise comment. The selected points in high vulnerability areas are selected based on MCDA output. These points are distributed in the city of Tehran at appropriate distances from each other so that there is at least one training example in each high vulnerability area.

Point 8: The neural network has a point prediction; how do the authors obtain the vulnerability values for the rest of the map? Has any type of interpolation been done?

Response:

Thanks for this comment. The input and output of the neural network model are not necessarily points. Neural network models are also flexible to implement on raster data. In this study, the criteria values for each pixel have been determined, by entering these values into the neural network, the degree of vulnerability is calculated at the pixel scale. Therefore, the neural network model can have a raster output.

Point 9:  Is there any reference to the vulnerability map of Tehran to compare the results reached by the authors and interpret their results? 

Response:

Many thanks for this precise comment. Yes, in the "Discussion" section, the results obtained in this study have been compared and discussed with the results of the previous study in the same area (Please see: pages 22 and 23, lines 505-542).

Round 2

Reviewer 1 Report

The manuscript has been significantly improved in terms of its organization and clarity. The author has addressed all my comments and concerns and responded reasonably to other reviewers' comments. I think the manuscript is in the publishable stage.

Reviewer 2 Report

Accept at present form